# Age of initiation of cigarillos, filtered cigars and/or traditional cigars among youth: Findings from the Population Assessment of Tobacco and Health (PATH) study, 2013–2017

**Baojiang Chen**[1,2], **Kymberle L. Sterling**[3], **Meagan A. Bluestein**[2], **Arnold E. Kuk**[2], **Melissa B. Harrell**[2,4], **Cheryl L. Perry**[2,3], **Adriana Pérez** [1,2]*

1 Department of Biostatistics and Data Science, School of Public Health, The University of Texas Health Science Center at Houston (UTHealth), Austin, Texas, United States of America, 2 Michael & Susan Dell Center for Healthy Living, School of Public Health, The University of Texas Health Science Center at Houston (UTHealth), Austin, Texas, United States of America, 3 Department of Health Promotion and Behavioral Sciences, School of Public Health, The University of Texas Health Science Center at Houston (UTHealth), Austin, Texas, United States of America, 4 Department of Epidemiology, Human Genetics and Environmental Sciences, School of Public Health, The University of Texas Health Science Center at Houston (UTHealth), Austin, Texas, United States of America

* adriana.perez@uth.tmc.edu

## Abstract

### Significance

Early age of initiation of tobacco use is associated with sustained tobacco use and lower rates of smoking cessation. Although much is known about age of initiation of cigarette use, much less is known about the age of initiation of cigar product use among youth.

### Methods

Survival analyses of the Population Assessment of Tobacco and Health youth annual data-sets (ages 12–17) from 2013 to 2017 were conducted for any cigar product use, cigarillos or filtered cigars, and traditional cigars across four cigar use outcomes, age of initiation of: susceptibility to use, ever use, past 30-day use and "fairly regular" use. An interval censoring survival method was implemented to estimate the probability of each outcome for age of initiation of each cigar product overall. Differences in age of initiation by sex and race/ethnicity were assessed using weighted Cox proportional hazards models for interval-censored data.

### Results

For each outcome across the three cigar products, striking increases in the probability of initiation begin before 17 years old. For cigarillo or filtered cigars, males had a higher risk of onset of susceptibility to use, initiating ever use, and initiating past 30-day use at earlier ages than females. Compared to Non-Hispanic Whites, Hispanic and Non-Hispanic Other had lower risk of initiating ever use and past 30-day use at earlier ages. Non-Hispanic Black youth had higher risk of initiating past 30-day use and "fairly regular" use than Non-Hispanic

**Data Availability Statement:** All the data from waves 1-4 are available from the Population

Assessment of Tobacco and Health (PATH) Study [United States] Restricted-Use Files. Inter-university Consortium for Political and Social Research [distributor], 2020-06-24. https://doi.org/10.3886/ICPSR36231.v25. Sample from PLOS ONE publication: Data are available from https://www.icpsr.umich.edu/icpsrweb/NAHDAP/studies/36498/datadocumentation#.

**Funding:** Research reported in this publication was supported by grant number [1R01CA234205-01A1] from the National Cancer Institute (NCI) and the FDA Center for Tobacco Products (https://projectreporter.nih.gov/project_info_description.cfm?aid=9928017&icde=51197168) to Dr. Adriana Perez. The content is solely the responsibility of the authors and does not necessarily represent the official views of the National Institutes of Health (NIH) or the Food and Drug Administration (FDA).

**Competing interests:** The authors have no conflicts of interest to disclose except Dr. Harrell is a consultant in litigation involving the vaping industry. This does not alter our adherence to PLOS ONE policies on sharing data and materials.

White youth at earlier ages. Similar findings are reported for any cigar use and traditional cigar use.

## Conclusion

Developmentally and culturally appropriate cigar use interventions and communication campaigns should be provided to youth before 17 years of age to prevent the onset and progression of cigar products. Regulatory policies that reduce appeal of all cigar products should be implemented to curb cigar initiation among youth.

## Introduction

According to the 2019 National Youth Tobacco Survey (NYTS) data, cigar products are the most popular combustible tobacco product used among high school (7.6%) and middle school (2.3%) youth, surpassing cigarettes (of which 5.8% of high and 2.3% of middle school students reported using) [1]. In addition, the 2019 NYTS data indicate that boys were more likely to use cigar products than girls [2]. Further, cigar products were the most commonly used tobacco product among Non-Hispanic Black high school students, with 9.2% (95% CI: 6.8–12.4%) reporting past 30-day use compared to other racial/ethnic groups (Non-Hispanic White: 7.8%, 95% CI: 6.7–9.1%; Hispanic: 7.3%, %95 CI: 5.9–9.1) [2]. Although the 2019 NYTS reported data for all cigar products, three types of cigars are sold in the United States: traditional cigars (large cigars or "regular" cigars that are ≥ 7 inches and contain 5–20 grams of tobacco), cigarillos (smaller cigars that are bigger than a cigarette and contain 3 grams of tobacco), and filtered cigars (same size and shape of cigarettes, maintaining a filter and contain 1 gram of tobacco) [3, 4]. Some youth smoke cigarillos as 'blunts' by removing the tobacco from its casing and replacing it with marijuana and still using the cigarillos or filtered cigars with some or none of the tobacco that originally came in these tobacco products [5–7]. In addition to differing by size, recent studies suggest that youth users have different patterns of use for these products. Frequently endorsed reasons for cigarillo and filtered cigar use among adolescents include curiosity, appealing flavors, friends' use, low cost [8] and believing that flavored cigar products are easier to use than non-flavored cigar products [7]. However, socializing when smoking traditional cigars was reported in PATH 2013–2014 as the most popular reason for traditional cigar use among adults [9]. While in 2013–2014, an analysis of cigar smoking patterns among adults indicated that filtered cigars are used as substitutes for cigarettes [9], it is unknown if this is also true in youth. The previous study also reported that cigarillos and filtered cigars had similar tobacco product characteristics and users had similar purchasing behavior, while the product characteristics and purchasing behaviors were different with traditional cigars [9]. For instance, 83% of adults reported having a regular brand for cigarillos and filtered cigars, while only 49% of adults reported having a brand for traditional cigars [9]. In the same study, the majority of adults reported a box or pack as the usual purchase size for cigarillos and filtered cigars while the majority of adults reported a single cigar as the usual purchase size for traditional cigars [9]. From the health perspective, cigarillos or filtered cigars are generally used more frequently in youth and are often inhaled, which may increase young smokers' risk for addiction to nicotine and/or poor health outcomes [10, 11]. However, the traditional cigars are usually not inhaled and are used less frequently in youth. Understanding the age at which youth become susceptible to, initiate and progress to established cigar product use and how the timing of initiation may vary for each cigar product and across demographic

characteristics can inform the development of prevention interventions and regulatory policies that reduce cigar products appeal among youth.

Data from the Population Assessment of Tobacco and Health (PATH) study, a longitudinal study of tobacco use and its effects on health in the United States [12], reported the percentage of cigar use initiation across a two year-interval (at wave 2 or wave 3) among wave 1 never-cigar using youth [13]. However, their cigar product use included any use of cigarillos, filtered cigars or traditional cigars. Additionally, among 12-17-year-old youth never users at wave 1, 9% (95% CI: 8.3–9.7%) initiated any cigar product use in the past 12-months; 4% (95% CI: 3.7–4.4%) initiated any cigar product use within the past 30-days, and 0.1% (95% CI: 0.1–0.2%) initiated frequent use of any cigar product within the past 30 days at wave 2 (2014–15) or wave 3 (2015–16) [13]. While this study of PATH provides important evidence about the proportion of cigar use onset among youth, findings for each cigar product type (e.g., cigarillos) were not provided. Also, the proportions were reported for age groups, instead of a specific age in years (e.g., 16-year old) [13]. Other studies have reported the recalled age of initiation of cigar use among adult ever users [14]. A previous study in 2018 reported that the recalled age of cigar product initiation was 13.6 years old among a sample of 9,889 ever cigar-using college students in Virginia Commonwealth University [14]. Although these findings shed light on the importance of early adolescence as a potential developmental risk period for cigar use initiation, recalled age of initiation is subject to recall bias [15, 16].

In this article, we present analyses from the first four waves of the PATH study that show the prospective distributions of age of initiation for different outcomes and cigar products. In contrast to previous cigar use initiation studies, we prospectively estimate the age of initiation across four years of data with interval-censored survival analyses for the combined 'any cigar product' category and each cigar product type, including cigarillos or filtered cigars and traditional cigars. Age of initiation across the four behavioral outcomes of susceptibility to use, ever use, past 30-day use, and "fairly regular" use are reported. We estimate the age at which youth initiated use for the overall sample and by sex and racial/ethnic group, thus providing a fine-tuned estimate of the age that youth initiate each cigar product category across four behavioral smoking outcomes. We go beyond prior studies by providing the field with evidence that will shed light on a more precise age of initiation for each cigar product, and the proportion of youth who initiated. These data will guide prevention interventions among youth and high-light the specific age, sex and race/ethnicity groups to target with communication and education campaigns for intervention.

## Methods

### Study design and participants

Secondary analyses of the PATH youth data were conducted. PATH is a longitudinal study of tobacco use and its effects on health in the United States that targets the population of individuals aged 12 and older in all 50 U.S. states, using a four-stage stratified sampling design annually from 2013–2014 (wave 1) to 2016–2017 (wave 4) and biannually thereafter. Full details of the study design are available with a summary presented here [17]. Four waves of PATH data are available to researchers (wave 2: 2014–2015, wave 3: 2015–2016, wave 4: 2016–2017). At wave 1, 13,651 youth (aged 12–17) completed the survey with a 78.4% response rate [18]. In addition, family members of PATH participants who were 9–11 years old at wave 1 were eligible and invited to participate in PATH when they reached 12 years of age at waves 2–3 (known as "shadow youth"), with 2,091 and 2,045 12 year old youth joining PATH in waves 2 and 3, respectively [19]. When youth turned 18 years old, they were invited to continue with the PATH adult measurements. There were 1,915, 1,907, and 1,900 youth who entered the adult

PATH study in waves 2–4, respectively, as they turned 18 [19]. The response rate in PATH youth in waves 2–4 were: 87.3%, 83.3% and 79.5%, respectively [19]. The original PATH investigators obtained written informed consent from all participants ages 18 and older with youth respondent's ages 12 to 17 providing written assent and each youth's parent/legal guardian providing written consent [20, 21]. In addition, all data were de-identified in order to preserve participant anonymity [22]. IRB approval for this study was obtained from the Committee for the Protection of Human Subjects at the University of Texas Health Science Center at Houston with number HSC-SPH-17-0368.

## Measures

The PATH study assesses use of cigarillos, filtered cigars and traditional cigars. In this study, we examined three classes of cigar products. PATH created derived variables for cigar product use, and that blunt-only users were excluded from these variables (i.e., those who used cigar products with marijuana inside). However, participants who used both, cigar as intended/sold and as blunts are included in the analysis [22]. The three classes of cigar products are: (i) any cigar use (defined as use of cigarillos, filtered cigars or traditional cigars); (ii) combined cigarillo or filtered cigar use; and (iii) traditional cigar use only. Our analysis of a combined category of cigarillo or filtered cigar product use is consistent with other published studies [9, 23]. Because the proportions of initiation for cigarillos alone and filtered cigars alone were small [7], we increased the power of this study by combining both products. We estimated the age of initiation for these products across four behavioral outcomes: (a) susceptibility to use; (b) ever use; (c) past 30-day use; and (d) "fairly regular" use.

**Susceptibility to use.** In PATH wave 1, the following questions were used to measure susceptibility to cigarillo use: (i) "Have you ever been curious about smoking a cigarillo?", (ii) "Do you think that you will try a cigarillo soon?", and (iii) "Would you smoke a cigarillo if one of your best friends offered you one?" [19, 22]. Response options for the first question were "very curious", "somewhat curious", "a little curious", and "not at all curious". Response options for the next two questions were "definitely yes", "probably yes", "probably not" and "definitely not". Participants who answered "not at all curious" to the first question and "definitely not" to the next two questions were considered non-susceptible to cigarillo use. Participants who had any other combination of answers were considered susceptible to cigarillo use. Similarly, these three questions were available for filtered and traditional cigars. Participants who were considered as susceptible to either cigarillos, filtered cigars or traditional cigars, were categorized as susceptible to any cigar product. Participants who were considered susceptible to either cigarillos or filtered cigars were categorized as susceptible for the cigarillo or filtered cigars outcome. Measurement and classification of cigarillo, filtered cigar and traditional cigar use susceptibility is consistent with that used in other tobacco use studies [24, 25].

**Ever use.** In waves 1–4, PATH asked youth respondents "Have you ever smoked a traditional cigar, cigarillo, or filtered cigar, even one or two puffs?" [19, 22]. PATH then derived three variables in the restricted dataset describing (i) ever cigarillo, (ii) ever filtered cigar and (iii) ever traditional cigar use (response options: yes/no). Participants who reported "yes" to ever use of either cigarillos, filtered cigars or traditional cigars were categorized as any cigar product users. Participants who reported "yes" to ever use of either cigarillos or filtered cigars were categorized as ever users for the cigarillo or filtered cigars outcome. Participants who reported "yes" to ever use of traditional cigars were categorized as ever users of traditional cigars only. Although, PATH also asked youth responders "Have you ever smoked part or all of a cigar, cigarillo or filtered cigar with marijuana in it?", for the purposes of this study the

cigars users include both "blunt" and "non-blunt users", but not those who reported use of only blunts and not the cigars products as intended.

**Past 30-day use.** In waves 1–4, PATH had asked the participants "In the past 30 days, on how many days did you smoke a [cigar product]?", separately for cigarillos, filtered cigars and traditional cigars [19, 22]. PATH derived three variables that assessed if youth respondents had ever smoked these products in the past 30 days: (i) cigarillos, (ii) filtered cigars, or (iii) traditional cigar (response options: yes/no). With these variables, three variables were implemented for analyses: (i) past 30-day use of either cigarillos, filtered cigars or traditional cigars to represent past 30-day any cigar product use; (ii) past 30-day use of cigarillos or filtered cigars; and past 30-day users of traditional cigars only.

**Fairly regular use.** In waves 2–4, "fairly regular" use of cigarillos was measured by "Have you ever smoked a cigarillo fairly regularly?". Response options included "yes", "no", "don't know" or "refused". The "don't know" or "refused" responses were classified as missing. A similar question was asked for filtered and traditional cigar product use. Three variables identified participants who indicated: (i) "fairly regular" use of either cigarillos, filtered cigars or traditional cigars; (ii) "fairly regular" use of either cigarillos or filtered cigars; and (iii) "fairly regular" use of traditional cigars.

## Inclusion/Exclusion criteria

For each cigar product (any cigar, cigarillos or filtered cigar, and traditional cigar only), participants who were less than 18 years old at their first wave of participation of PATH study, were not susceptible or had not initiated use of any cigar product at study entry were included in the sample of non-susceptible and never users of each product to have their cigar use outcomes followed-up prospectively. Those who were non-susceptible were followed-up to determine age of first report of susceptibility, and those who were never users were followed-up to determine age of initiation of ever use, past 30-day use, and "fairly regular" use.

## Sex and race/Ethnicity

Sex was classified as males or females. This variable was imputed using the household information by PATH at wave 1. In PATH, race was assessed as White race alone, Black race alone, Asian race alone, and other race (including multi-racial), and ethnicity was categorized as either Hispanic or Non-Hispanic. In our analyses, to be comparable to those in prior Surgeon General's reports [23, 26] and other studies, we classify race/ethnicity into four categories: Non-Hispanic White, Hispanic, Non-Hispanic Black, Non-Hispanic Other (Asian, multi-race, and other races).

## Interval-censored outcome: age of initiation of cigars

The age of initiation of each cigar use behavior (susceptibility, ever, past 30-day, and fairly regular use) and each cigar product (i.e., any cigar, cigarillos or filtered cigars and traditional cigars) was prospectively estimated using the participant's age and the dates between surveys within one week of precision because PATH does not provide participants' date of birth in the restricted use file [18, 22]. Specifically, the age of initiation was estimated by adding the participants' age at their first wave of PATH participation to the number of weeks between subsequent waves based on when the cigar use behavior and product was first reported among youth who end up initiating or the last report of never/non-use among those who did not initiate. For each subject, a lower and upper bound was calculated, where the lower bound was the age at the last wave where the subject reported non-use of each cigar use behavior and product, and the upper bound was the age that the subject first reported use of each cigar use behavior and product. If the subject did not initiate cigar use, the upper bound was censored.

## Statistical analysis

All analyses conducted incorporate the sampling weights and the 100 balance repeated replicate weights to account for the complex survey design with Fay's factor of 0.3 in PATH [27]. All statistical analyses were completed in SAS version 9.4 using the Inter-university Consortium for Political and Social Research server hosted by the University of Michigan. Weighted statistics are reported, with means and standard errors for continuous variables and frequencies and percentages for categorical variables [18, 22]. The nonparametric survival analyses for interval-censored data were implemented to estimate the hazards of different types of cigar initiation overall, as well as stratified by sex and race/ethnicity [28, 29]. The hazard function of initiation of different cigar use behavior and products and its 95% confidence intervals (CI) are reported. The association between the age of initiation of different cigar use behaviors and products with sex and race/ethnicity were conducted using the Cox proportional hazards regression model with interval-censored data and the piecewise constant baseline hazard function [30]. The hazard ratio (HR) and its 95% CI are reported. Separate hazard functions showing the full distributions for the ages of initiation and their 95% CIs are also reported for the cigar use behaviors and products with significant effects. For all analyses, the estimate was represented as "NA" if there is not enough sample size to provide stable estimates at that age.

## Results

### Overview

In Table 1, we present the demographic characteristics of the youth in our analytic sample, stratified by non-susceptible and never users of any cigar product at their first wave of PATH participation. Additionally, we present the cumulative probabilities of the age of onset of susceptibility to use, and age of initiation of ever, past 30-day and "fairly regular" use for the overall sample (Table 2), by sex (Table 5) and by race/ethnicity (Table 3). The associations between sex and race/ethnicity and age of onset of susceptibility to use, and the age of initiation of ever, past 30-day and "fairly regular" use for any cigar, cigarillos or filtered cigars, and traditional cigars are presented in Table 4. Figs 1–3 also report the overall hazard functions of the ages of initiation for all four outcomes with the three cigar products, as well as stratified by sex and race/ethnicity.

The sections below describe in detail the age of initiation data for cigarillos or filtered cigars. We focus our presentation on cigarillos or filtered cigars as these are the most popular cigar products among youth. The sociodemographic characteristics of the sample are presented, followed by descriptions of the age of initiation data by behavioral outcomes (i.e., susceptibility to use, ever use, past 30-day use, and "fairly regular" use). A brief report is presented of findings for traditional cigar products.

### Sociodemographic characteristics of sample

Several samples of never cigar product users were utilized, given variation in cigar product use. As shown in Table 1, there were n = 7,873 youth, representing N = 14,977,381 participants who were classified as non-susceptible to and had never used cigarillos or filtered cigars at their first wave of entry into PATH between 2013–2016. Among these participants, almost 68% entered PATH in 2013–2014, 52% were female, 19% were Hispanic, 57% were Non-Hispanic White, 14% were Non-Hispanic Black, and 9% were Non-Hispanic other. Their mean age was about 14 years old. Table 1 presents the demographic characteristics of other non-susceptible, never users of any cigars (n = 7763, N = 14,748,765) and traditional cigars (n = 7858, N = 14,953,824).

**Table 1. Demographic characteristics of PATH USA youth (aged 12–17) non-susceptible or never cigar users at the first wave of study participation in 2013–2016[*].**

| Cigar products | Any Cigar use | | Cigarillos or filtered cigar use | | Traditional Cigar Use | |
|---|---|---|---|---|---|---|
| Sample at the first wave of study participation | Never Any Cigar Users (n = 16,363, N = 30,571,813) | Respondents Not Susceptible to Any Cigar (n = 7,763, N = 14,748,765) | Never Cigarillo or filtered cigar (n = 16,492, N = 30,817,710) | Respondents Not Susceptible to Cigarillos or filtered cigar (n = 7,873, N = 14,977,381) | Never Traditional cigar Users (n = 17,193, N = 32,045,150) | Respondents Not Susceptible to traditional cigar (n = 7,858, N = 14,953,824) |
| Variables | Weighted Frequency (%) | Weighted Frequency (%) | Weighted Frequency (%) | Weighted Frequency (%) | Weighted Frequency (%) | Weighted Frequency (%) |
| **Wave of Entry Into PATH** | | | | | | |
| Wave 1 | 22,339,969 (73.1) | 10,129,630 (68.7) | 22,572,567 (73.2) | 10,129,630 (67.6) | 23,786,447 (74.2) | 10,129,630 (67.7) |
| Wave 2 | 4,071,805 (13.3) | 2,306,739 (15.6) | 4,078,910 (13.2) | 2,418,358 (16.1) | 4,088,765 (12.8) | 2,397,901 (16.0) |
| Wave 3 | 4,160,038 (13.6) | 2,312,396 (15.7) | 4,166,232 (13.5) | 2,429,393 (16.2) | 4,169,938 (13.0) | 2,426,293 (16.2) |
| **Sex** | | | | | | |
| Female | 15,369,986 (50.3) | 7,677,980 (52.1) | 15,519,075 (50.4) | 7,781,473 (52.0) | 16,232,965 (50.7) | 7,784,967 (52.1) |
| Male | 15,181,878 (49.7) | 7,061,582 (47.9) | 15,278,686 (49.6) | 7,186,704 (48.0) | 15,792,236 (49.3) | 7,156,959 (47.9) |
| Missing | 19,949 | 9,203 | 19949 | 9,204 | 19,949 | 11,898 |
| **Race/ethnicity** | | | | | | |
| Hispanic | 7,084,355 (23.5) | 2,826,236 (19.4) | 7,115,196 (23.4) | 2,856,713 (19.3) | 7,417,338 (23.4) | 2,897,833 (19.6) |
| Non-Hispanic White | 16,019,101 (53.1) | 8,347,083 (57.3) | 16,199,667 (53.3) | 8,473,824 (57.3) | 16,780,448 (53.0) | 8,410,659 (57.0) |
| Non-Hispanic Black | 4,093,793 (13.6) | 2,041,009 (14.0) | 4,108,715 (13.5) | 2,071,183 (14.0) | 4,354,042 (13.8) | 2,070,506 (14.0) |
| on-Hispanic Other[†] | 2,977,509 (9.9) | 1,355,345 (9.3) | 2,994,842 (9.8) | 1,383,381 (9.4) | 3,096,268 (9.8) | 1,382,004 (9.4) |
| Missing | 397,055 | 179,092 | 399,290 | 192,280 | 397,054 | 192,822 |
| **Age at entry into study (SE)** | 13.8 (0.006) | 13.7 (0.018) | 13.8 (0.006) | 13.7 (0.018) | 13.9 (0.004) | 13.7 (0.017) |
| **# of Other Tobacco Products Used** | | | | | | |
| 0 | 26,936,110 (88.1) | 13,453,734 (91.2) | 27,014,595 (87.7) | 13,658,706 (91.2) | 27,091,580 (84.5) | 13,619,789 (91.1) |
| 1 | 2,224,708 (7.3) | 915,954 (6.2) | 2,280,739 (7.4) | 927,354 (6.2) | 2,384,630 (7.4) | 931,225 (6.2) |
| 2 | 951,331 (3.1) | 292,882 (2.0) | 994,801 (3.2) | 298,185 (2.0) | 1,165,067 (3.6) | 303,725 (2.0) |
| 3+ | 459,663 (1.5) | 82,776 (0.6) | 527,575 (1.7) | 89,716 (0.6) | 1,403,873 (4.5) | 92,970 (0.7) |

[*]PATH Restricted file received disclosure to publish: February 10, 2020, February 24, 2020, March 09, 2020, March 25, 2020, May 19. 2020 and October 2, 2020. United States Department of Health and Human Services. National Institutes of Health. National Institute on Drug Abuse, and United States Department of Health and Human Services. Food and Drug Administration. Center for Tobacco Products. Population Assessment of Tobacco and Health (PATH) Study [United States] Restricted-Use Files. ICPSR 36231-v13.AnnArbor, MI: Inter-university Consortium for Political and Social Research [distributor], November 5, 2019. Https://doi.org/10.3886/ICPSR36231.v23.

† Non-Hispanic Others include Asian, multi-race, etc.

There were n = 16,492 youth, representing N = 30,817,710 participants in PATH, who reported never use of cigarillos or little filtered cigars at their first wave of entry into the PATH study between 2013–2016. Among these, 73% entered the study in 2013–2014, 50% were female, 23% were Hispanic, 53% were Non-Hispanic White, 14% were Non-Hispanic Black, and 10% were Non-Hispanic Other (Including Asian, multi-race, etc.). Their mean age was 13.8. Other samples of analysis included those for any cigars (n = 16,363, N = 30,571,813) and traditional cigars (n = 17,193, N = 32,045,150), whose demographic characteristics are reported in Table 1.

**Table 2. Estimated hazard function[a] (and 95% confidence intervals)[b] of the age of initiation of cigar outcomes for the overall sample of PATH USA[*].**

| Age | Susceptibility (%) | Ever use (%) | Past 30-day use (%) | Fairly regular use (%) |
|---|---|---|---|---|
| **Any Cigar Use** | | | | |
| **15** | 15.2 (14.2–16.2) | 1.5 (1.3–1.8) | 0.6 (0.5–0.8) | 0.6 (0.4–0.9) |
| **16** | 22.4 (21.1–23.6) | 3.3 (2.9–3.7) | 1.2 (1.0–1.5) | 1.3 (0.9–1.7) |
| **17** | 31.4 (30.0–32.8) | 6.3 (5.7–7.0) | 2.3 (1.9–2.7) | 2.0 (1.4–2.5) |
| **17.25** | 42.8 (39.6–46.1) | 11.1 (8.4–13.8) | 6.2 (4.1–8.4) | NA |
| **17.5** | 42.8 (39.7–46.0) | 14.6 (12.2–17.1) | 8.6 (6.6–10.5) | NA |
| **17.75** | 43.1 (40.2–46.0) | 15.3 (13.5–17.0) | 9.9 (8.1–11.7) | NA |
| **18** | 47.4 (43.6–51.3) | 21.1 (18.3–23.8) | 11.3 (9.8–12.9) | NA |
| **19** | 54.3 (51.3–57.4) | 28.6 (26.2–30.9) | 17.6 (15.9–19.3) | NA |
| **20** | NA | 31 (28.7–33.3) | 20.9 (18.9–23.0) | NA |
| **Cigarillo or Filtered Cigar Use** | | | | |
| **15** | 11.9 (11.2–12.7) | 1.4 (1.1–1.6) | 0.5 (0.4–0.7) | 0.6 (0.4–0.9) |
| **16** | 17.4 (16.4–18.4) | 2.9 (2.5–3.2) | 1.1 (0.8–1.3) | 1.2 (0.8–1.6) |
| **17** | 24.7 (23.4–26.0) | 5.5 (4.9–6.1) | 1.9 (1.6–2.3) | 2 (1.6–2.6) |
| **17.25** | 31.5 (28.4–34.7) | 9.3 (6.3–12.3) | 5.9 (3.4–8.4) | 45.4 (37.5–53.4)ⱡ |
| **17.5** | 31.8 (27.9–35.6) | 11.8 (10.1–13.6) | 7.1 (5.4–8.8) | 45.4 (37.5–53.4)ⱡ |
| **17.75** | 34.3 (31.4–37.2) | 12.9 (11.3–14.6) | 8.5 (6.7–10.3) | 45.4 (37.5–53.4)ⱡ |
| **18** | 38.0 (35.2–40.7) | 18.0 (15.1–20.8) | 10.1 (8.2–11.9) | 45.4 (37.5–53.4)ⱡ |
| **19** | 44.9 (41.8–47.9) | 23.9 (21.4–26.3) | 15.4 (13.9–17.0) | 53.6 (46.4–60.8)ⱡ |
| **20** | NA | 26.5 (24.3–28.8) | 18.4 (16.6–20.3) | 53.6 (46.4–60.8)ⱡ |
| **Traditional Cigar Use** | | | | |
| **15** | 10.1 (9.4–10.8) | 0.4 (0.3–0.6) | 0.2 (0.1–0.3) | NA |
| **16** | 14.6 (13.6–15.6) | 1.1 (0.9–1.3) | 0.5 (0.3–0.6) | NA |
| **17** | 21.3 (20.1–22.6) | 2.3 (1.9–2.6) | 0.8 (0.6–1.0) | NA |
| **17.25** | 26.5 (23.5–29.5) | 5.7 (3.9–7.6) | 1.9 (1.2–2.6) | NA |
| **17.5** | 26.5 (23.6–29.4) | 7.2 (4.8–9.7) | 3.3 (1.7–4.8) | NA |
| **17.75** | 29.0 (26.4–31.6) | 8.1 (5.9–10.3) | 4.8 (3.4–6.3) | NA |
| **18** | 31.3 (28.9–33.7) | 12.8 (10.9–14.6) | 5.9 (5.1–6.8) | NA |
| **19** | 37.9 (34.9–40.9) | 19.5 (17.7–21.3) | 8.9 (7.7–10.1) | NA |
| **20** | NA | 21.4 (19.2–23.6) | 11.7 (10.0–13.3) | NA |

[a]: Hazards are reported as cumulative percentages (i.e., cumulative incidence).

[b]: 95% CI: Turnbull 95% confidence interval.

[*]PATH Restricted file received disclosure to publish: February 10, 2020, February 24, 2020, March 09, 2020, March 25, 2020, May 19. 2020. United States Department of Health and Human Services. National Institutes of Health. National Institute on Drug Abuse, and United States Department of Health and Human Services. Food and Drug Administration. Center for Tobacco Products. Population Assessment of Tobacco and Health (PATH) Study [United States] Restricted-Use Files. ICPSR 36231-v13.AnnArbor, MI: Inter-university Consortium for Political and Social Research [distributor], November 5, 2019. Https://doi.org/10.3886/ICPSR36231.v23.

NA: there was not enough sample size to produce a stable probability estimate at this age.

ⱡ The estimate is not reliable due to smaller sample sizes.

## Age of onset of susceptibility to cigarillos or filtered cigars use

As shown in Table 2, among youth who were non-susceptible to cigarillo or filtered cigar use at their first wave of PATH participation, 11.9% became susceptible to cigarillo or filtered cigar use by age 15. Between ages 16 and 17 years old, the probability for becoming susceptible to cigarillo or filtered cigar use increased by 7.3% (17.4% to 24.7%), while a 13.3% increase in susceptibility risk occurred between ages 17 and 18 years (24.7% to 38%). As shown in Table 5, males exhibited a higher risk for becoming susceptible to cigarillo or filtered cigar use within

**Table 3. Estimated hazard function[a] (and 95% confidence intervals)[b] of age of initiation of cigar outcomes for the overall sample of PATH USA youth (aged 12–17) by race/ethnicity[*].**

| Age | Non-Hispanic White | Hispanic | Non-Hispanic Black | Non-Hispanic Other[†] |
|---|---|---|---|---|
| **Susceptibility of Traditional Cigar** | | | | |
| 15 | 8.6 (7.7–9.6) | 11.5 (9.8–13.3) | 12 (9.8–14.2) | 12.1 (9.1–15.1) |
| 16 | 13.1 (11.9–14.3) | 16.8 (14.4–19.3) | 17 (14.2–19.8) | 15.8 (12.3–19.3) |
| 17 | 19.9 (18.2–21.7) | 23.3 (20.3–26.2) | 23.4 (19.5–17.3) | 22.6 (18.3–26.8) |
| 17.25 | 27.8 (24.4–31.2) | 27.3 (23.0–31.5) | 28.1 (21.1–35.2) | 22.6 (18.3–26.9) |
| 17.5 | 27.8 (24.8–30.8) | 27.3 (23.0–31.5) | 28.1 (21.1–35.2) | 22.6 (18.3–26.9) |
| 17.75 | 27.8 (25.0–30.6) | 30.6 (25.1–36.1) | 32.0 (25.8–38.2) | 22.6 (18.3–26.9) |
| 18 | 29.8 (16.5–33.1) | 34.6 (30.4–38.8) | 32.9 (26.9–39.0) | 37.4 (29.2–45.7) |
| 19 | 37.1 (32.4–41.8) | 43.9 (34.8–53.0) | 36.8 (30.2–43.4) | 40.5 (25.9–55.2) |
| **Ever Use of Any Cigar** | | | | |
| 15 | 1.5(1.2–1.8) | 1.7(1.2–2.2) | 1.8(0.9–2.6) | 1.2(0.5–1.9) |
| 16 | 3.6(3.0–4.2) | 2.6(2.0–3.2) | 4(2.7–5.4) | 2.2(1.3–3.1) |
| 17 | 7.1(6.1–8.1) | 4.9(3.9–6.0) | 7.3(5.5–9.0) | 4.5(2.8–6.1) |
| 17.25 | 11.3 (8.6–14.0) | 7.1 (5.3–9.0) | 10.6 (6.2–14.9) | 8.0 (4.0–12.0) |
| 17.5 | 17.0 (13.9–20.1) | 12.0 (9.2–14.7) | 14.0 (6.7–21.2) | 8.0 (3.9–12.0) |
| 17.75 | 17.0 (14.5–19.5) | 12.0 (9.4–14.5) | 15.0 (10.7–19.3) | 14.5 (9.5–19.6) |
| 18 | 23.2(19.7–26.7) | 16.2(12.5–19.8) | 19.5(15.3–23.7) | 15.1(10.6–19.5) |
| 19 | 29.8(26.6–32.9) | 21.3(17.1–25.5) | 34(28.4–39.5) | 23.2(18.5–28.0) |
| 20 | 33.7(30.5–36.9) | 27.3(22.7–30.7) | 34(28.5–39.4) | 23.2(18.5–28.0) |
| **Ever Use of Cigarillo or Filtered Cigar** | | | | |
| 15 | 1.2 (1.0–1.5) | 1.4 (0.9–1.9) | 1.7 (0.9–2.6) | 1.3 (0.5–2.1) |
| 16 | 3 (2.5–3.6) | 2.2 (1.6–2.8) | 4 (2.6–5.4) | 2.3 (1.2–3.3) |
| 17 | 6 (5.0–7.0) | 4.2 (3.2–5.2) | 7.3 (5.5–9.0) | 4 (2.6–5.4) |
| 17.25 | 10.1 (7.5–12.7) | 6.2 (4.1–8.3) | 10.6 (6.2–15.0) | 6.9 (3.7–10.2) |
| 17.5 | 13.4 (11.0–15.8) | 9.9 (7.4–12.4) | 13.6 (6.3–21.0) | 6.9 (3.7–10.2) |
| 17.75 | 13.4 (11.0–15.8) | 9.9 (7.6–12.2) | 15.8 (11.5–20.0) | 12.3 (7.9–16.8) |
| 18 | 19.1 (15.3–22.8) | 14 (10.8–17.1) | 18.3 (14.2–22.4) | 12.9 (8.3–17.5) |
| 19 | 24 (20.6–27.5) | 17.6 (14.0–21.2) | 31.8 (26.5–37.2) | 20.7 (16.1–25.2) |
| 20 | 28.4 (25.2–31.6) | 23.5 (19.6–27.5) | 31.8 (26.2–37.5) | 20.7 (16.1–25.2) |
| **Ever Use of Traditional Cigar** | | | | |
| 15 | 0.5 (0.3–0.6) | 0.5 (0.2–0.8) | 0.2 (0.0–0.5) | 0.4 (0.0–0.8) |
| 16 | 1.3 (1.0–1.7) | 0.8 (0.4–1.2) | 0.8 (0.2–1.3) | 1.3 (0.6–2.0) |
| 17 | 2.9 (2.3–3.4) | 1.8 (1.2–2.3) | 1.2 (0.7–1.8) | 1.9 (0.9–3.0) |
| 17.25 | 6.6 (3.4–9.8) | 4.1 (1.6–6.6) | 1.3 (0.2–2.4) | 5.0 (1.6–8.4) |
| 17.5 | 10.5 (7.0–14.0) | 5.5 (2.6–8.4) | 1.3 (0.2–2.4) | 5.0 (1.6–8.4) |
| 17.75 | 10.8 (8.1–13.5) | 5.5 (3.1–7.8) | 3.6 (1.5–5.7) | 8.8 (4.4–13.2) |
| 18 | 16.6 (14.2–19.1) | 8.3 (5.3–11.4) | 5.9 (3.3–8.4) | 12.2 (7.1–16.6) |
| 19 | 24.2 (21.6–26.9) | 12.3 (8.6–15.9) | 14.3 (10.2–18.5) | 15.3 (11.2–19.4) |
| 20 | 26.3 (23.3–29.4) | 15.5 (12.3–18.8) | 14.3 (10.7–18.0) | 15.3 (11.2–19.4) |
| **Past 30-Day Use of Any Cigar** | | | | |
| 15 | 0.5 (0.3–0.8) | 0.7 (0.4–1.0) | 1.0 (0.4–1.6) | 0.5 (0.0–1.0) |
| 16 | 1.3 (0.9–1.7) | 0.9 (0.5–1.3) | 1.7 (0.9–2.6) | 1.1 (0.2–2.0) |
| 17 | 2.8 (2.1–3.4) | 1.6 (1.0–2.1) | 2.8 (1.8–3.9) | 1.4 (0.6–2.2) |
| 17.25 | 5.3 (3.3–7.3) | 3.4 (1.1–5.7) | 7.9 (3.6–12.3) | 4.3 (1.6–7.0) |
| 17.5 | 9.7 (7.3–12.1) | 6.1 (4.7–7.5) | 7.9 (3.3–12.6) | 4.3 (1.6–7.0) |
| 17.75 | 9.7 (6.9–12.4) | 6.1 (4.7–7.5) | 11.0 (7.4–14.6) | 8.5 (4.0–13.1) |

*(Continued)*

**Table 3.** (Continued)

| Age | Non-Hispanic White | Hispanic | Non-Hispanic Black | Non-Hispanic Other[†] |
|---|---|---|---|---|
| **18** | 12.3 (10.7–14.0) | 8.2 (5.3–11.2) | 13.2 (9.9–16.6) | 9.5 (6.2–12.8) |
| **19** | 18.1 (15.9–20.4) | 12.9 (10.3–15.5) | 24.1 (17.8–30.5) | 13.8 (10.1–17.5) |
| **20** | 21.2 (18.3–24.2) | 19.0 (14.8–23.3) | 28.2 (22.1–34.2) | 13.8 (10.1–17.5) |
| **Past 30-Day Use of Cigarillo or Filtered Cigar** | | | | |
| **15** | 0.5 (0.3–0.7) | 0.5 (0.2–0.8) | 0.9 (0.3–1.5) | 0.6 (0.1–1.1) |
| **16** | 1.1 (0.8–1.4) | 0.7 (0.3–1.1) | 1.6 (0.8–2.5) | 1.1 (0.3–1.9) |
| **17** | 2.2 (1.6–2.7) | 1.4 (0.8–1.9) | 2.6 (1.5–3.7) | 1.5 (0.8–2.2) |
| **17.25** | 4.9 (3.0–6.8) | 2.8 (0.3–5.4) | 7.2 (3.3–11.1) | 4.8 (2.3–7.3) |
| **17.5** | 7.4 (5.5–9.2) | 5.3 (3.9–6.7) | 7.2 (2.3–12.1) | 4.8 (2.3–7.3) |
| **17.75** | 7.4 (4.9–9.9) | 5.3 (3.9–6.7) | 11.4 (7.7–15.1) | 7.3 (3.6–11.0) |
| **18** | 9.8 (7.4–12.2) | 7 (4.6–9.3) | 12.8 (9.3–16.3) | 7.5 (4.8–10.3) |
| **19** | 15.8 (13.8–17.8) | 11.1 (.2–13.0) | 23.6 (17.1–30.2) | 12.3 (8.7–15.9) |
| **20** | 18.2 (15.8–20.6) | 16.3 (12.5–20.1) | 28.5 (22.3–34.7) | 12.3 (8.7–15.9) |
| **Past 30-Day Use Traditional Cigar** | | | | |
| **15** | 0.2 (0.1–0.3) | 0.2 (0.1–0.4) | 0.1 (0.0–0.4) | 0.3 (0.0–0.7) |
| **16** | 0.6 (0.3–0.8) | 0.3 (0.1–0.5) | 0.4 (0.1–0.8) | 0.3 (0.0–0.7) |
| **17** | 1.2 (0.8–1.6) | 0.4 (0.1–0.8) | 0.6 (0.2–1.0) | 0.5 (0.1–0.9) |
| **17.25** | 2.2 (1.3–3.2) | 1.6 (0.3–2.9) | 0.9 (0.0–2.0) | 1.6 (0.0–3.4) |
| **17.5** | 4.6 (1.7–7.6) | 2.5 (1.3–3.6) | 0.9 (0.0–2.0) | 1.6 (0.0–3.4) |
| **17.75** | 6.6 (4.7–8.4) | 2.5 (1.5–3.4) | 2.7 (1.1–4.3) | 3.7 (1.6–5.8) |
| **18** | 8 (6.7–9.3) | 3.3 (1.8–4.8) | 3.4 (1.7–5.1) | 6.2 (3.5–8.8) |
| **19** | 11.1 (9.3–12.9) | 6.6 (4.9–8.4) | 6.5 (3.3–9.7) | 7.2 (3.8–10.6) |
| **20** | 13.8 (11.4–16.1) | 10.1 (6.6–13.6) | 7.9 (5.4–10.4) | 7.9 (5.0–10.8) |
| **Fairly Regular Use of Cigarillo or Filtered Cigar** | | | | |
| **15** | 0.6 (0.3–1.0) | 0.7 (0.2–1.2) | 0.6 (0.0–1.3) | 0.6 (0.0–1.1) |
| **16** | 1.3 (0.8–1.8) | 0.7 (0.2–1.2) | 2.6 (1.0–4.3) | 0.6 (0.0–1.3) |
| **17** | 2 (1.2–2.9) | 1.1 (0.4–1.8) | 2.7 (1.0–4.4) | 2.6 (0.6–4.6) |
| **17.25** | 36.5 (27.5–45.6) | 27.4 (0.0–64.2) | 71.8 (43.8–99.3) | 2.6 (0.6–4.6) |
| **17.5** | 36.5 (27.5–45.6) | 52.5 (9.9–95.1) | 71.8 (43.8–99.3) | 2.6 (0.6–4.6) |
| **17.75** | 36.5 (27.5–45.6) | 53.5 (22.3–85.7) | 71.8 (49.6–93.9) | 52.5 (16.5–88.4) |
| **18** | 36.5 (27.8–45.2) | 53.6 (36.3–70.5) | 78.5 (63.4–93.5) | 52.5 (20.2–84.7) |
| **19** | 44.7 (33.8–55.7) | 53.6 (36.3–70.5) | 78.5 (63.4–93.5) | 52.5 (20.2–84.7) |
| **20** | 44.7 (33.8–55.7) | 53.6 (36.3–70.5) | 78.5 (63.4–93.5) | NA |

[a]: Hazards are reported as cumulative percentages (i.e., cumulative incidence).

[b]: 95% CI: Turnbull 95% confidence interval.

[*]PATH Restricted file received disclosure to publish: February 10, 2020, February 24, 2020, March 09, 2020, March 25, 2020, May 19. 2020. United States Department of Health and Human Services. National Institutes of Health. National Institute on Drug Abuse, and United States Department of Health and Human Services. Food and Drug Administration. Center for Tobacco Products. Population Assessment of Tobacco and Health (PATH) Study [United States] Restricted-Use Files. ICPSR 36231-v13.AnnArbor, MI: Inter-university Consortium for Political and Social Research [distributor], November 5, 2019. Https://doi.org/10.3886/ICPSR36231.v23.

NA: there was not enough sample size to produce a stable probability estimate at this age.

[†]Non-Hispanic Others include Asian, multi-race, etc.

the latter time frame, with 15.1% becoming susceptible to use from ages 17 to 18 (25.9% to 41%), compared to 12% of females from 17 to 18 years (23.7% to 35.7%). Overall, males were 11% (HR: 1.11, 95% CI: 1.01–1.22) more likely to be susceptible to cigarillo or filtered cigar use at a younger age than females (Table 4). There were no significant differences for the age of initiation for susceptibility to cigarillo or filtered cigar use by race/ethnicity (Table 4).

**Table 4. Hazard ratio (and 95% confidence intervals) for each cigar product initiation outcome by sex and race/ethnicity[*].**

| Variables | Susceptibility to | Ever use | Past 30-day Use | Fairly regular use |
|---|---|---|---|---|
| **Any Cigar Use** | | | | |
| **Sex** | | | | |
| Female | 1.00 | 1.00 | 1.00 | 1.00 |
| Male | **1.28 (1.18–1.39)** | **1.72 (1.55–1.92)** | **1.98 (1.72–2.26)** | 1.59 (0.96–2.63) |
| **Race** | | | | |
| Non-Hispanic White | 1.00 | 1.00 | 1.00 | 1.00 |
| Non-Hispanic Black | 0.99 (0.87–1.13) | 1.00 (0.82–1.22) | **1.26 (1.0–1.59)** | 1.50 (0.79–2.85) |
| Non-Hispanic Other[†] | 1.03 (0.84–1.27) | **0.65 (0.51–0.82)** | **0.66 (0.50–0.87)** | 1.00 (0.46–2.14) |
| Hispanic | 1.07 (0.94–1.21) | **0.68 (0.60–0.78)** | **0.72 (0.61–0.85)** | 0.62 (0.33–1.16) |
| **Cigarillo or Filtered Cigar Use** | | | | |
| **Sex** | | | | |
| Female | 1.00 | 1.00 | 1.00 | 1.00 |
| Male | **1.11 (1.01–1.22)** | **1.61 (1.43–1.82)** | **1.79 (1.54–2.04)** | 0.92 (0.69–1.22) |
| **Race** | | | | |
| Non-Hispanic White | 1.00 | 1.00 | 1.00 | 1.00 |
| Non-Hispanic Black | 1.11 (0.95–1.29) | 1.18 (0.95–1.46) | **1.47 (1.17–1.86)** | **2.11 (1.36–3.27)** |
| Non-Hispanic Other[†] | 1.10 (0.88–1.37) | **0.70 (0.56–0.89)** | **0.70 (0.53–0.92)** | 1.42 (0.89–2.27) |
| Hispanic | 1.12 (0.99–1.27) | **0.72 (0.62–0.83)** | **0.72 (0.60–0.86)** | 1.00 (0.64–1.54) |
| **Traditional Cigar Use** | | | | |
| **Sex** | | | | |
| Female | 1.00 | 1.00 | 1.00 | 1.00 |
| Male | 1.10 (0.99–1.22) | **3.03 (2.56–3.45)** | **3.33 (2.63–4.35)** | NA |
| **Race** | | | | |
| Non-Hispanic White | 1.00 | 1.00 | 1.00 | 1.00 |
| Non-Hispanic Black | 1.18 (0.98–1.41) | **0.43 (0.33–0.55)** | **0.49 (0.35–0.69)** | NA |
| Non-Hispanic Other[†] | 1.11 (0.89–1.39) | **0.60 (0.45–0.80)** | **0.59 (0.42–0.84)** | NA |
| Hispanic | **1.21 (1.05–1.41)** | **0.53 (0.44–0.63)** | **0.56(0.45–0.70)** | NA |

[a]: Hazards are reported as cumulative percentages (i.e., cumulative incidence).

[b]: 95% CI: Turnbull 95% confidence interval.

[*]PATH Restricted file received disclosure to publish: February 10, 2020, February 24, 2020, March 09, 2020, March 25, 2020, May 19. 2020. United States Department of Health and Human Services. National Institutes of Health. National Institute on Drug Abuse, and United States Department of Health and Human Services. Food and Drug Administration. Center for Tobacco Products. Population Assessment of Tobacco and Health (PATH) Study [United States] Restricted-Use Files. ICPSR 36231-v13.AnnArbor, MI: Inter-university Consortium for Political and Social Research [distributor], November 5, 2019. Https://doi.org/10.3886/ICPSR36231.v23.

NA: there was not enough sample size to produce a stable probability estimate at this age

†Non-Hispanic Others include Asian, multi-race, etc.

## Age of initiation of ever use of cigarillos or filtered cigars

While the hazard of becoming an ever cigarillo or filtered cigar user was low before age 15, striking increases in the probability of initiation of ever use occurred as youth become older. As shown in Table 2, between ages 17 and 18 years, we observed a 12.5% increase (5.5% to 18%) in those who initiated ever cigarillo or filtered cigar use. Table 5 shows that males exhibited a heightened risk for onset of ever use at earlier ages, with 9% initiating ever use from ages 17 to 17.5 (5.9% to 14.9%) and 6.2% initiating ever use from 17.5 to 18 years (14.9% to 21.1%). For females, 4.4% initiated ever use from ages 17 to 17.5 (5.3% to 9.7%) and 3.4% initiated ever use from 17.5 to 18 years (9.7% to 13.1%). Overall, males were 61% (HR: 1.61, 95% CI: 1.43–1.82) more likely to initiate ever cigarillo or filtered cigar use at a younger age than females (see

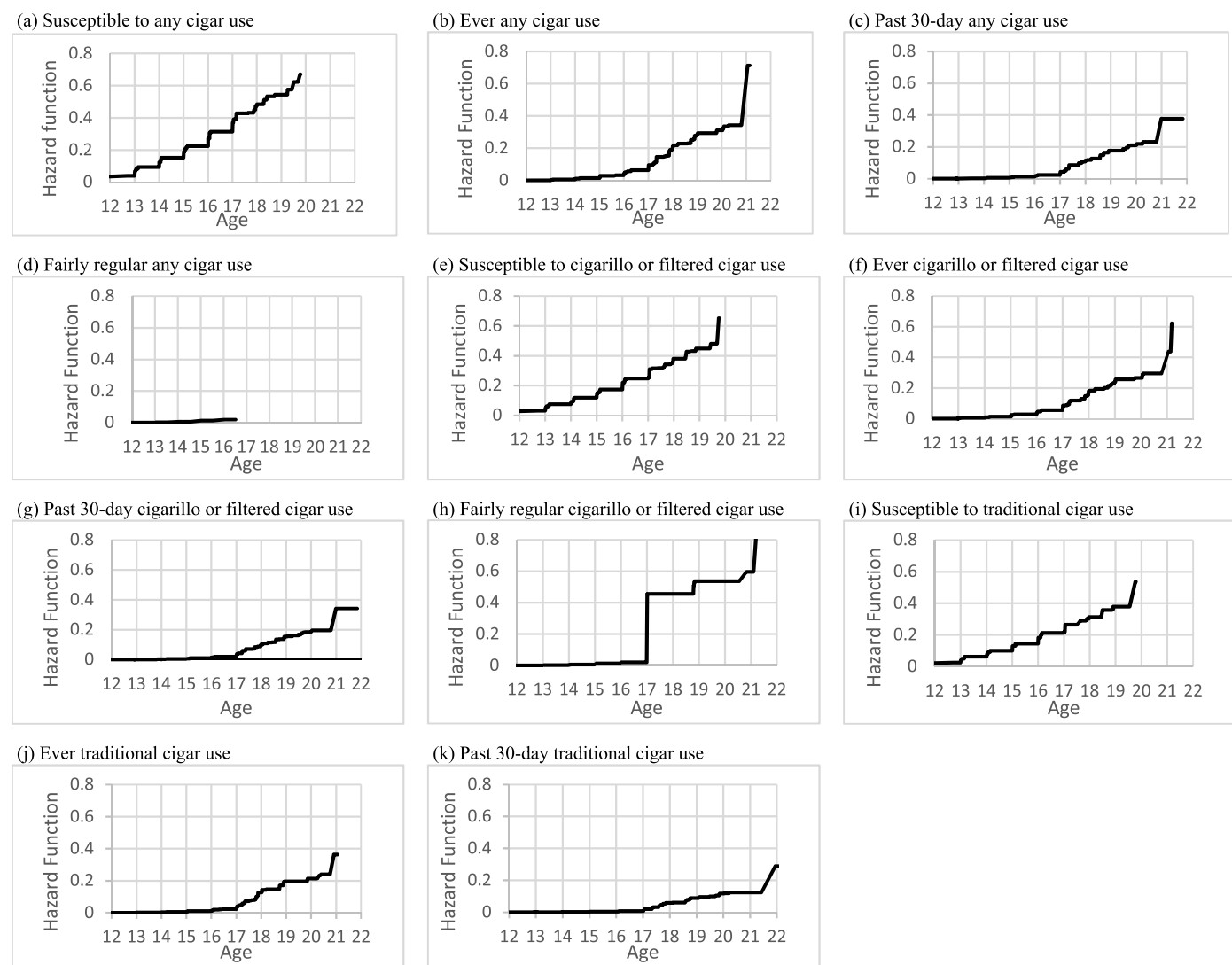

**Fig 1. Estimated hazard function for age of initiation by cigar use type.**

Table 4). As shown in Table 4, youth who identified as Hispanic or Non-Hispanic Other (e.g. Asian, multi-race, etc.) were less likely to initiate cigarillo or filtered cigar use at earlier ages than those who were identified as Non-Hispanic White (HR of Hispanic vs Non-Hispanic White: 0.72, 95% CI: 0.62–0.83; HR of Non-Hispanic Other vs Non-Hispanic White: 0.70, 95% CI: 0.56–0.89).

## Age of initiation of past 30-day use of cigarillos or filtered cigars

As shown in Table 2, marked increases in the hazard of initiating past 30-day cigarillo or filtered cigar use occurred between ages 17 and 18, where an 8.2% increase was observed (1.9% to 10.1%). Among males, initiation of past 30-day cigarillo or filtered cigar use increased by 9.8% (2.2% to 12.0%) between ages 17 and 18 (Table 5). Between ages 17 to 17.5, a 7% increase in initiation of past 30-day use was observed for males (2.2% to 9.2%). Initiation of past 30-day use increased among males by an additional 2.8% between ages 17.5 and 18 years (9.2% to 12.0%). For females, between ages 17 and 17.5, a 3.2% increase in initiation of past 30-day use was observed (1.7% to 4.9%), and between ages 17.5 and 18, a 2.2% increase in initiation of past

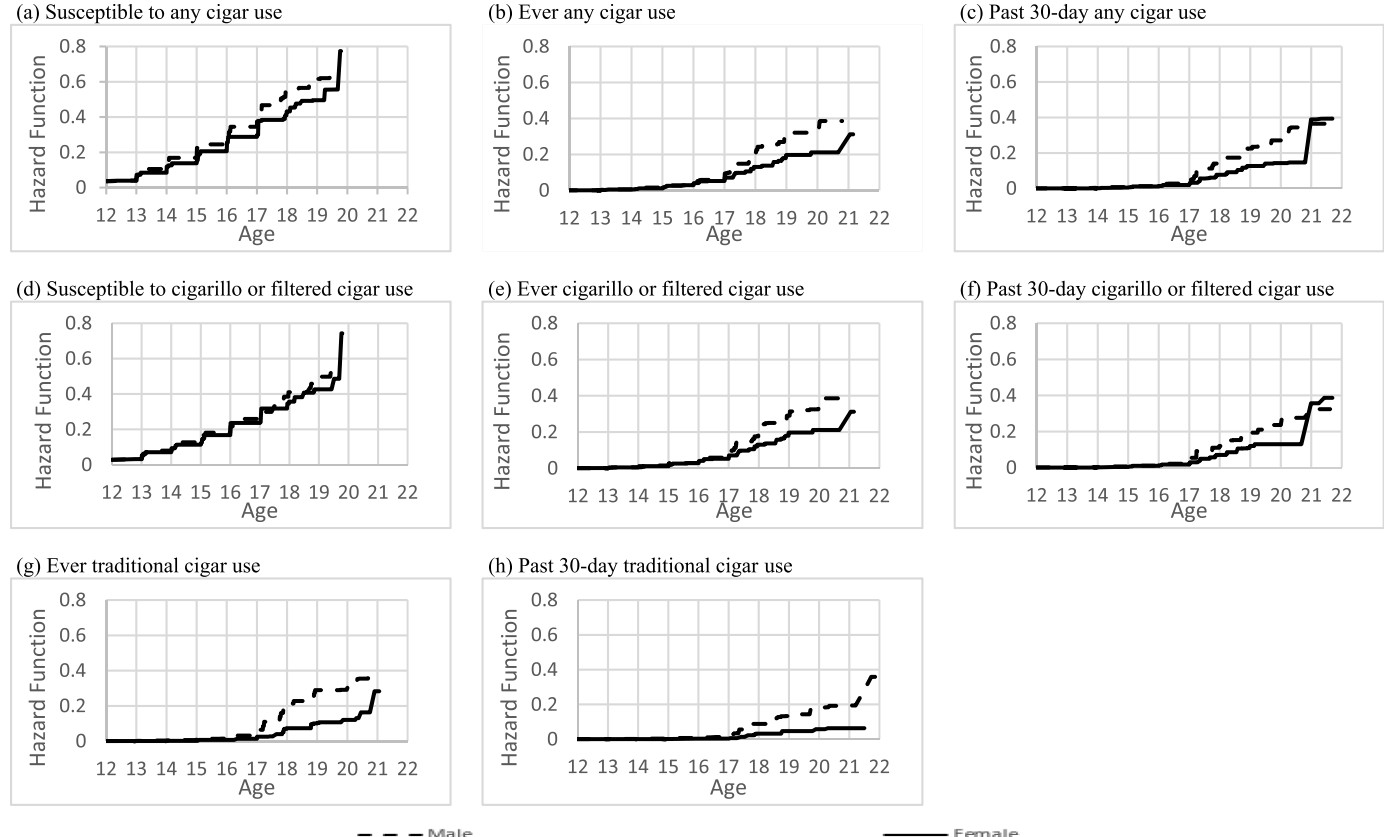

**Fig 2. Estimated hazard function for age of initiation of different types of cigar use, stratified by sex: Males are the dotted line and females are the solid line.**

30-day use was observed (4.9% to 7.1%). Overall, males were 79% (HR: 1.79, 95% CI: 1.54–2.04) more likely to experience the initiation of past 30-day use at a younger age than females (Table 4). As shown in Table 4, compared to Non-Hispanic Whites, youth who were identified as Non-Hispanic Black were 47% (HR: 1.47, 95% CI: 1.17–1.86) more likely to initiate past 30-day cigarillo or filtered cigar users at younger ages. Youth who were identified as Non-Hispanic Other were 30% (HR: 0.70, 95% CI: 0.53–0.92) less likely to be past 30-day cigarillo or filtered cigar users at younger ages compared to Non-Hispanic White youth. Youth who were identified as Hispanic were 28% (HR: 0.72, 95% CI: 0.60–0.86) less likely to be past 30-day cigarillo or filtered cigar users at younger ages compared to Non-Hispanic White youth.

## Age of initiation of "fairly regular" use of cigarillos or filtered cigars

The hazard of becoming a "fairly regular" cigarillo or filtered cigar user was low, resulting in 2% initiation by age 17. As indicated in Table 4, there was no significant difference for the initiation of "fairly regular" cigarillo or filtered cigar use between males and females. Youth who were identified as Non-Hispanic Black were 111% (HR: 2.11, 95% CI: 1.36–3.27) more likely to be "fairly regular" cigarillo or filtered cigar users at earlier ages than those were identified as Non-Hispanic White.

## Age of onset of susceptibility to traditional cigar use

As shown in Table 2, among youth who were non-susceptible to traditional cigar use at their first wave of PATH participation, 10.1% became susceptible to traditional cigar use by age 15.

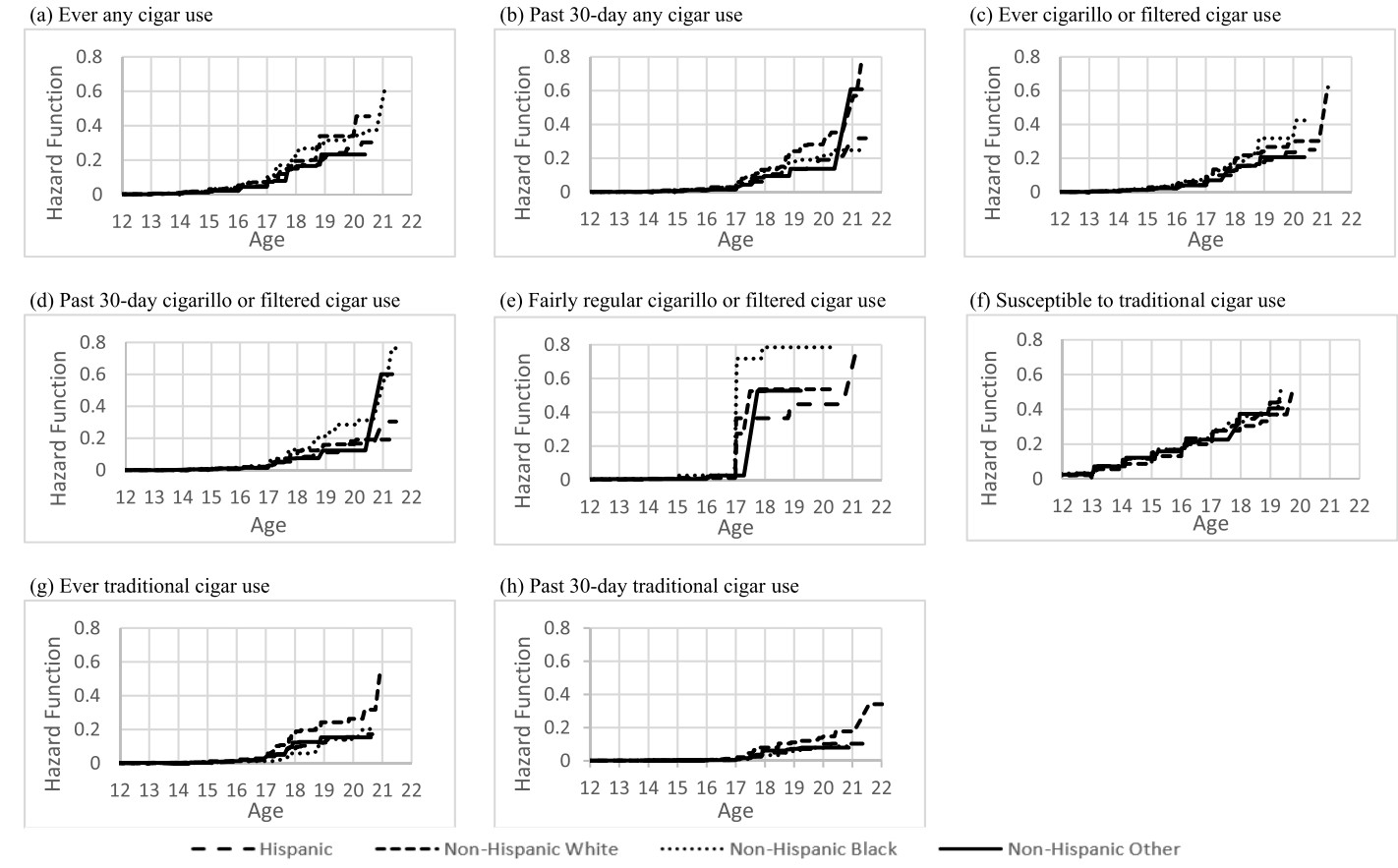

**Fig 3. Estimated hazard function for age of initiation of different types of cigar use stratified by race and ethnicity.**

Between ages 16 and 17 years old, the probability of becoming susceptible to traditional cigar use increased by 6.7% (14.6% to 21.3%), while a 10% increase in susceptibility risk occurred between ages 17 and 18 years (21.3% to 31.3%). As shown in Table 4, Hispanic youth had higher risk to be susceptible to traditional cigar use than Non-Hispanic White youth (HR = 1.21, 95% CI: 1.05–1.41). There were no significant differences for the age of onset of susceptibility to traditional cigar use by sex (Table 4).

## Age of initiation of ever use of traditional cigars

As shown in Table 2, the hazard of initiating an ever-traditional cigar user was low before age 17. Between ages 17 and 18 years, we observed a 10.5% increase (2.3% to 12.8%) in those who initiated ever traditional cigar use. Table 5 shows that males exhibited a heightened risk for initiating ever use within this time frame, with 15.4% initiating ever use from ages 17 to 18 (3.2% to 18.6%). For females, 6.1% initiated ever use from ages 17 to 18 (1.3% to 7.4%). Overall, males were 203% (HR: 3.03, 95% CI: 2.56–3.45) more likely to be ever users at a younger age than females (see Table 4). As shown in Table 4, youth who were identified as Non-Hispanic Black, Hispanic or Non-Hispanic Other were less likely to initiate traditional cigar use at earlier ages than those who were identified as Non-Hispanic White (HR of Non-Hispanic Black vs Non-Hispanic White: 0.43, 95% CI: 0.33–0.55; HR of Hispanic vs Non-Hispanic White: 0.53, 95% CI: 0.44–0.63; HR of Non-Hispanic Other vs Non-Hispanic White: 0.60, 95% CI: 0.45–0.80).

**Table 5. Estimated hazards[a] (and 95% confidence intervals)[b] of age of initiation of each cigar outcomes for the overall sample of PATH USA youth (aged 12–17) by sex[*].**

| Age | Susceptibility | | Ever use | | Past 30-day use | |
|---|---|---|---|---|---|---|
| | Male | Female | Male | Female | Male | Female |
| **Any Cigar use** | | | | | | |
| 15 | 16.8 (15.3–18.4) | 13.8 (12.7–15.0) | 1.8 (1.4–2.2) | 1.3 (1.1–1.6) | 0.8 (0.5–1.1) | 0.5 (0.3–0.7) |
| 16 | 24.5 (22.5–26.5) | 20.6 (19.1–22.0) | 3.3 (2.8–3.9) | 3.2 (2.7–3.8) | 1.3 (1.0–1.7) | 1.1 (0.8–1.4) |
| 17 | 34.5 (32.3–36.7) | 28.8 (27.0–30.6) | 6.9 (5.9–7.9) | 5.7 (4.9–6.6) | 2.8 (2.1–3.4) | 1.9 (1.5–2.4) |
| 17.25 | 46.7 (42.2–51.1) | 38.4 (35.1–41.6) | 16.8 (11.3–22.3) | 7.5 (6.3–8.8) | 10.6 (6.5–14.6) | 3.2 (2.2–4.2) |
| 17.5 | 46.7 (42.1–51.3) | 38.4 (35.4–41.4) | 18.5 (15.0–22.0) | 11.1 (8.9–13.3) | 11.3 (8.8–13.7) | 5.6 (4.7–7.6) |
| 17.75 | 46.7 (41.9–51.4) | 38.4 (35.5–41.2) | 18.5 (15.6–21.3) | 11.4 (9.4–13.4) | 11.3 (8.1–14.4) | 6.1 (4.7–7.5) |
| 18 | 54.4 (49.3–59.5) | 43.2 (38.6–47.7) | 24 (19.8–28.2) | 15.4 (13.5–17.3) | 14.0 (11.7–16.3) | 7.8 (6.4–9.1) |
| 19 | 61.6 (54.7–68.4) | 49.6 (45.8–53.3) | 35.2 (30.7–39.8) | 21.8 (19.5–24.0) | 22.5 (20.0–25.1) | 12.7 (10.8–14.6) |
| 20 | NA | NA | 37.2 (34.2–40.3) | 24.6 (22.0–27.2) | 27.1 (23.7–30.4) | 14.3 (12.4–16.3) |
| **Cigarillo or filtered cigar use** | | | | | | |
| 15 | 12.3 (11.6–13.9) | 11.3 (10.2–12.4) | 1.6 (1.2–1.9) | 1.2 (0.9–1.4) | 0.7 (0.4–0.9) | 0.4 (0.2–0.6) |
| 16 | 18.1 (16.6–19.6) | 16.7 (15.4–18.1) | 2.9 (2.4–3.4) | 2.9 (2.4–3.4) | 1.1 (0.8–1.4) | 1 (0.7–1.3) |
| 17 | 25.9 (24.1–27.7) | 23.7 (22.1–25.3) | 5.9 (4.9–6.8) | 5.3 (4.5–6.1) | 2.2 (1.6–2.8) | 1.7 (1.3–2.1) |
| 17.25 | 29.9 (25.7–34.2) | 31.8 (29.2–34.4) | 12.9 (8.4–17.4) | 7.1 (5.8–8.4) | 9.2 (5.2–13.2) | 3.0 (1.9–4.1) |
| 17.5 | 31.1 (25.7–40.6) | 31.8 (29.3–34.4) | 14.9 (11.8–17.9) | 9.7 (7.6–11.9) | 9.2 (6.8–11.6) | 4.9 (3.0–6.8) |
| 17.75 | 35.2 (30.2–40.1) | 31.8 (29.4–34.3) | 14.9 (11.8–17.9) | 10.5 (8.6–12.5) | 9.2 (6.7–11.7) | 5.7 (4.3–7.1) |
| 18 | 41 (36.6–45.4) | 35.7 (32.1–39.2) | 21.1 (17.0–25.4) | 13.1 (11.4–14.8) | 12 (9.5–14.6) | 7.1 (5.8–8.4) |
| 19 | 48.4 (42.8–53.9) | 42.7 (39.4–46.0) | 29 (25.0–33.1) | 19.6 (16.5–22.8) | 19.4 (17.0–21.8) | 11.9 (9.6–14.1) |
| 20 | NA | NA | 32.5 (29.3–35.8) | 21.2 (18.7–23.6) | 23.6 (20.3–26.9) | 13 (11.2–14.8) |
| **Traditional cigar use** | | | | | | |
| 15 | 10.6 (9.5–11.7) | 9.7 (8.7–10.7) | 0.6 (0.4–0.9) | 0.2 (0.1–0.3) | 0.3 (0.1–0.5) | 0.1 (0.0–0.2) |
| 16 | 15.3 (13.9–16.8) | 14.0 (12.8–15.3) | 1.5 (1.1–1.8) | 0.7 (0.5–1.0) | 0.7 (0.4–0.9) | 0.2 (0.1–0.3) |
| 17 | 22.5 (20.9–24.2) | 20.3 (18.7–21.9) | 3.2 (2.6–3.8) | 1.3 (1.0–1.7) | 1.3 (0.9–1.7) | 0.4 (0.2–0.6) |
| 17.25 | 26.0 (22.5–29.6) | 27.1 (24.6–29.7) | 11.0 (6.5–15.6) | 2.5 (1.6–3.4) | 3.4 (1.6–5.2) | 0.6 (0.2–1.1) |
| 17.5 | 26.0 (21.6–30.5) | 27.1 (24.6–29.7) | 11.3 (8.3–14.3) | 2.7 (1.1–4.4) | 5.5 (3.1–8.0) | 1.3 (0.0–2.9) |
| 17.75 | 29.6 (25.4–33.9) | 27.1 (24.6–29.7) | 11.5 (7.8–15.3) | 4.0 (2.7–5.3) | 6.5 (4.3–8.6) | 2.3 (1.4–3.1) |
| 18 | 33.1 (29.5–36.6) | 29.6 (26.3–32.8) | 18.6 (16.0–21.2) | 7.4 (5.9–8.8) | 8.2 (7.5–10.2) | 3.2 (2.4–4.1) |
| 19 | 41.4 (35.8–47.0) | 35.7 (32.2–39.2) | 29 (26.0–32.0) | 10.1 (8.5–11.7) | 13.3 (11.1–15.5) | 4.6 (3.5–5.7) |
| 20 | NA | NA | 29.1 (26.2–32.0) | 12 (10.0–14.1) | 17.4 (14.7–20.0) | 5.8 (4.0–7.6) |

[a]: Hazards are reported as cumulative percentages (i.e., cumulative incidence).

[b]: 95% CI: Turnbull 95% confidence interval.

[*]PATH Restricted file received disclosure to publish: February 10, 2020, February 24, 2020, March 09, 2020, March 25, 2020, May 19. 2020. United States Department of Health and Human Services. National Institutes of Health. National Institute on Drug Abuse, and United States Department of Health and Human Services. Food and Drug Administration. Center for Tobacco Products. Population Assessment of Tobacco and Health (PATH) Study [United States] Restricted-Use Files. ICPSR 36231-v13.AnnArbor, MI: Inter-university Consortium for Political and Social Research [distributor], November 5, 2019. Https://doi.org/10.3886/ICPSR36231.v23.

NA: there was not enough sample size to produce a stable probability estimate at this age.

## Age of initiation of past 30-day use of traditional cigars

As shown in Table 2, marked increases in the hazard of initiating past 30-day traditional cigar use occurred between ages 17 and 18, where a 5.1% increase was observed (0.8% to 5.9%). Among males, initiation of past 30-day traditional cigar use increased by 6.9% (1.3% to 8.2%) between ages 17 and 18 (Table 5). For females, between ages 17 and 18, a 2.8% increase in initiation of past 30-day use was observed (0.4% to 3.2%). Overall, males were 233% (HR: 3.33, 95% CI: 2.63–4.35) more likely to experience the initiation of past 30-day use at a younger age

than females (Table 4). As shown in Table 4, compared to Non-Hispanic Whites, youth who were identified as Non-Hispanic Black were 51% (HR: 0.49, 95% CI: 0.35–0.69) less likely to initiate past 30-day traditional cigar users at younger ages. Youth who identified as Non-Hispanic Other were 41% (HR: 0.59, 95% CI: 0.42–0.84) less likely to initiate past 30-day traditional cigar users at younger ages compared to Non-Hispanic White youth. Youth who were identified as Hispanic were 44% (HR: 0.56, 95% CI: 0.45–0.70) less likely to initiate past 30-day traditional cigar users at younger ages compared to Non-Hispanic White youth.

## Discussion

To our knowledge, our study is the first in the field to use survival analysis to estimate the distribution of the age of onset of susceptibility to cigar use and age of initiation of ever, past 30-day and "fairly regular" cigar use among youth never cigar users. Using data from four waves of the PATH study (2013–2017), we estimated the age of initiation for each cigar product category and behavioral outcome within a week's precision, using the participant's age and the number of weeks between survey dates. Our data provide the field with precise "windows of opportunity" to offer developmentally and culturally appropriate interventions to prevent the onset and progression of cigar product use among youth.

Tobacco product susceptibility is a gateway behavior and can be a useful indicator to assess progression to future use [31]. While prior studies have reported the prevalence of cigar use susceptibility among youth (ranging from 11.5% - 35.9% depending on sample characteristics [2, 32–34]), to our knowledge this is the first study to estimate an age by which youth first report onset of susceptibility of cigarillo or filtered cigar, as well as traditional cigar use. Between ages 15 and 18, we found a 26% increase (11.9% to 38%) in the proportion of youth who became susceptible to using cigarillos or filtered cigars and a 21% increase (10.1% to 31.3%) in those who became susceptible to use traditional cigars. By age 18, 18% of youth progressed to initiating ever use of cigarillos or filtered cigars, and 10% had progressed to initiating past 30-day use. Similar trends were found for traditional cigar use. Increases in the probability of initiation of ever and past 30-day cigarillo or filtered cigars and traditional cigar use continued to rise as these youth continued their transition into young adulthood. The incremental increases in risk of initiating ever and past 30-day cigarillo or filtered cigar or traditional cigar use may be related to increased access to freedom and financial resources that happens during young adulthood [35], resulting in increased experimentation and use with tobacco products. Cigarillos or filtered cigars and traditional cigars are often sold in appealing flavors [36, 37] and are advertised at point-of-sale in convenience stores and bodegas where young people shop [9, 38, 39]. In an analysis of PATH study data, any receptivity to cigar product advertising at wave 1 (2013–2014) was associated with cigar use progression (although this study defines progression as susceptibility or ever use) at wave 2 (2014–2015, OR = 2.01, 95% CI: 1.62–2.49) and with cigar product use (OR = 2.07; 95% CI: 1.26–3.40) [40]. Interventions that lower young people's receptivity to cigar product advertisements, and our findings indicate these should be provided to youth before early to mid-adolescence to be effective in preventing susceptibility to and progression of use. It is important to note that the age of initiation of cigarillos or filtered cigars occurred at earlier ages than traditional cigar use. Regulatory policies that limit cigar product advertisement at point-of-sale also will aid in reducing exposure to these advertisements, thereby preventing susceptibility and use progression. Prevention campaigns for youth should start for cigarillos and filtered cigars at earlier ages (i.e., before 15–16 years old), while campaigns for traditional cigars can be initiated before 16–17 years of age.

Our study supports prior studies that report sex and racial/ethnic differences in cigar product use. In the 2019 NYTS data, 16.9% males reported ever use of any cigar product, in

comparison to 11.8% for females [2]. Additionally, in our study, we found that males were more likely to become susceptible to and initiate ever and past 30-day cigar use than females at earlier ages. The 2019 NYTS data also indicate that cigar products are the most commonly used tobacco product among Non-Hispanic Black youth [2]. Our study provides additional evidence that supports findings of racial/ethnic disparities in cigar product use. Non-Hispanic Black youth were more likely to initiate past 30-day cigarillo or filtered cigar use and less likely to initiate ever and past 30-day traditional cigar use at earlier ages compared to Non-Hispanic White youth. Although data are limited for youth, previous studies of Non-Hispanic Black young adults report overwhelmingly positive attitudes toward cigarillos or filtered cigars and ubiquitous use in their communities–which may contribute to cigar initiation [41]. Cigar products are often used as a carrier for marijuana and smoked as a blunt [5]. This practice is considered safer than cigarette use among some Non-Hispanic Black youth, which could contribute to racial/ethnic differences in cigar product initiation [5, 42–44]. With regard to traditional cigars, Hispanic youth were more likely to be susceptible to use but less likely to initiate ever and past 30-day traditional cigar use at earlier ages compared to Non-Hispanic Whites. Non-Hispanic Black youth and those who were identified as Non-Hispanic "Other" were less likely to initiative ever and past 30-day traditional cigar use at earlier ages than Non-Hispanic White. Our findings indicate that race/ethnicity can serve as both a facilitator and protective factor for the age of initiation of certain cigar product types. Two prior studies among adult users found that there were differences in use behaviors (e.g., frequency, purpose) and product characteristics (e.g., price, flavorings, packaging size, accessibility, etc.) between traditional cigars, cigarillos and little filtered cigars [9, 45], as well as by race/ethnicity and top 5 brands [46]. Although those findings pertain to adults, it is possible that differences in use behaviors and product characteristics may influence racial/ethnic disparities in youth cigar product use. Future studies are needed to examine the complex interactions among sex, race/ethnicity and age of initiation for each cigar product.

"Fairly regular" use was also assessed for cigarillos or filtered cigars, and very few youths reported initiation of "fairly regular" use of these products. It is plausible that youth cigarillo or filtered cigar smoking behavior had not yet progressed to "fairly regular" use. It is also plausible that the measurement of "fairly regular" use is problematic with respect to cigarillo or filtered cigar product use. Unlike cigarettes, qualitative studies of cigarillo or filtered cigar smokers indicate that users only consume a portion of the product, modify it or use it socially [5, 42]. Additionally, traditional cigars are often used occasionally. It is possible that the measurement and operationalization of "fairly regular" use is not congruent with cigarillo, filtered cigar, or cigar smoking behavior. Future studies may consider using cognitive interviews or other techniques to understand the applicability of the operationalization of "fairly regular" use to cigarillo or filtered cigar users.

Strengths of the study include the use of a national (PATH) dataset, using the precise measurement of the age of initiation across four waves of PATH data (within a week's precision), the use of the prospective design which avoids the recalled bias, and the use of the Cox proportional hazards model for the interval-censored data. The Cox proportional hazard model describes the data more precisely than traditional statistical methods of measuring incidence or prevalence. However, the study is not without limitations. There were subjects who were already susceptible or had initiated some cigar outcomes (e.g., ever use, past 30-day use, and "fairly regular" use) at the first wave of participation into PATH, leading to left-censoring subjects and they were excluded in this analysis. Because our goal was to prospectively estimate the distribution of the age of initiation of these cigar product outcomes from non-susceptible or never users, we started with those who were non-susceptible or never users of these products at the first wave of participation into PATH to study how the cigar use behaviors emerge over time.

Users of little filtered cigars or cigarillos may use the product with marijuana as a blunt [6]. Although the PATH study includes a measure that asks if respondents have ever smoked part or all of a cigar, cigarillo or filtered cigar with marijuana in it, we did not use this variable in the study and age of initiation of blunt only users was not assessed. While we excluded "blunt-only" users, it is important to note that those cigar, cigarillo or filtered cigar users who may be dual using with blunts or who have had a history of blunt use were included in the sample. Additional research is needed to estimate and distinguish the age of initiation of cigar, cigarillo or filtered cigar among blunt only users versus those who have used the products without marijuana (i.e., non-blunt users). Our study starts with never users of cigar products but it is possible that never users of cigar products may have used cigarettes or other tobacco products at the first wave of participation into PATH or started those in concurrent use with cigar products. Future analyses of the age of initiation of cigar products should explore the influence of cigarette or other tobacco product use before or concurrent with cigar product initiation.

In conclusion, our study provides the field with evidence about the age of initiation of any cigars, cigarillos or filtered cigars and traditional cigars across several smoking behavior outcomes. Our study findings have important implications for future intervention research and tobacco regulatory science policy. While some health communication campaigns, such as the FDA Center for Tobacco Products' Fresh Empire campaign, have been developed to prevent and reduce tobacco use (including cigar use) among at risk youth age 12–17 years old who identified with hip-hop culture, to our knowledge no other prevention interventions or cessation treatment programs exist specifically geared to prevent the initiation of cigarillo or filtered cigar use or traditional cigars among youth or by sex or by race/ethnicity. Typically, cigarette prevention interventions are "repurposed" to focus on cigar products, which may not be sufficient to prevent the onset or continuation of use. Our data indicate that developmentally and culturally appropriate interventions need to be developed to prevent progression of cigarillo or filtered cigar and traditional cigar use behaviors–especially among youth from vulnerable populations who often experience tobacco-related health disparities. Interventions should also target important parent-child-family and peer-to-peer relationships, which have been documented as important factors to delay onset of tobacco product use [46–51]. In addition to cigarettes and ENDS use, health care providers should also screen for cigar product use among youth and specifically ask about cigarillo, filtered cigar or traditional cigar product use. Regarding tobacco regulatory policy, our findings indicate the importance of continuing to develop policies that reduce the appeal of all cigar products among youth.

## Author Contributions

**Conceptualization:** Baojiang Chen, Meagan A. Bluestein, Cheryl L. Perry, Adriana Pérez.

**Formal analysis:** Baojiang Chen, Adriana Pérez.

**Funding acquisition:** Meagan A. Bluestein, Cheryl L. Perry, Adriana Pérez.

**Investigation:** Arnold E. Kuk, Adriana Pérez.

**Methodology:** Adriana Pérez.

**Software:** Baojiang Chen, Adriana Pérez.

**Supervision:** Baojiang Chen, Adriana Pérez.

**Visualization:** Adriana Pérez.

**Writing – original draft:** Baojiang Chen, Kymberle L. Sterling, Adriana Pérez.

**Writing – review & editing:** Baojiang Chen, Kymberle L. Sterling, Meagan A. Bluestein, Arnold E. Kuk, Melissa B. Harrell, Cheryl L. Perry, Adriana Pérez.

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
