## [Decision Letter · Decision Letter 0]

11 Sep 2020

PONE-D-20-26733

AGE OF INITIATION OF CIGARILLOS, FILTERED CIGARS AND/OR TRADITIONAL CIGARS AMONG YOUTH: FINDINGS FROM THE POPULATION ASSESSMENT OF TOBACCO AND HEALTH STUDY, 2013-2017

PLOS ONE

Dear Dr. Perez,

Thank you for submitting your manuscript to PLOS ONE. After careful consideration, we feel the subject of your paper has merit but that the paper and analyses presented does not fully meet PLOS ONE’s publication criteria as it currently stands. Therefore, we invite you to consider a revised version of the manuscript that addresses the many excellent comments raised by our external reviewers. 

We look forward to receiving your revised manuscript.

Kind regards,

Michael Cummings, PhD

Academic Editor

PLOS ONE

Journal Requirements:

For additional information about PLOS ONE ethical requirements for human subjects research, please refer to " ext-link-type="uri" xlink:type="simple">http://journals.plos.org/plosone/s/submission-guidelines#loc-human-subjects-research."

"The authors have no conflicts of interest to disclose except Dr. Harrell is a consultant in

litigation involving the vaping industry"

Reviewers' comments:

Reviewer's Responses to Questions

**Comments to the Author**

1. Is the manuscript technically sound, and do the data support the conclusions?

Reviewer #1: Partly

Reviewer #2: Yes

2. Has the statistical analysis been performed appropriately and rigorously? 

Reviewer #1: Yes

Reviewer #2: Yes

3. Have the authors made all data underlying the findings in their manuscript fully available?

Reviewer #1: Yes

Reviewer #2: No

4. Is the manuscript presented in an intelligible fashion and written in standard English?

Reviewer #1: Yes

Reviewer #2: Yes

5. Review Comments to the Author

Reviewer #1: AGE OF INITIATION OF CIGARILLOS, FILTERED CIGARS AND/OR TRADITIONAL CIGARS AMONG YOUTH: FINDINGS FROM THE POPULATION ASSESSMENT OF TOBACCO AND HEALTH STUDY, 2013-2017

This study used data from PATH to investigate the age of initiation of cigars among youth. The study found that cigar use starts early in the teenage years similar to cigarette use.

Page 7, line 118. Because of demographic differences in use of cigars as blunts (Cullen, J., Mowery, P., Delnevo, C., Allen, J. A., Sokol, N., Byron, M. J. and Thornton-Bullock, A. Seven-year patterns in US cigar use epidemiology among young adults aged 18-25 years: a focus on race/ethnicity and brand. Am J Public Health. 2011 Oct;101(10):1955-62. doi: 10.2105/AJPH.2011.300209. Epub 2011 Aug 18.), it is important to be clear how these subjects were treated. How specifically did the authors exclude blunt use? The PATH question is “Have you ever smoked part or all of a cigar, cigarillo or filtered cigar with marijuana in it?” Did the authors exclude all blunt users even if they had also been cigar users or did they approach this in another way? Please clarify this question for susceptibility, ever use, past 30-day use, and fairly regular use.

Page 7, line 120. The authors state that other studies combined cigarillo and filtered cigar use. But, why, for this specific study, was it necessary and advantageous to combine cigarillo and filtered cigar use? The demographics of use of these two classes of cigars are quite different. Cigarillos are widely used as blunts and filtered cigars are used as substitutes for cigarettes. It is also possible that age of initiation may be different. Were there data that suggested that these data from cigarillo and filtered cigars were equivalent and could be readily combined in the context of this study’s objectives?

Page 8, line 140 and following. The discussion of methods for ever use, past 30-day use, and fairly regular use are very repetitious. These can be combined by listing the three PATH questions and then explaining how the participants were categorized based on their responses without repeating these steps three times.

Page 9, line 173. How did the authors address dual use? Were dual users included or excluded? Were these exclusive cigar users or were cigar/cigarette dual users also included in the test population? If so, how many of each category were excluded due to dual users?

Page 12, line 230. Table 2 and Figures 1-3 are sowing the same data. Either the Tables or the Figures should be in Supplemental instead.

Page 20, line 333. But there may have been differences by race/ethnicity if the authors had analyzed cigarillo and filtered cigars separately.

Page 25, line 448. Since age of initiation of traditional cigar use is much higher than for cigarillo and filtered cigars (manuscript lines 389-390), what does that suggest for FDA regulation of these products? Should they be regulated differently? If FDA were to use a youth-protection rationale for regulating cigars, should they include traditional cigars in the regulation?

Page 26, line 460. Here the authors describe the reason they should be clear about their rationale and methods for excluding blunt users in this study.

Page 27, line 473. The authors need to discuss possible differences in the meaning of “fairly regular” between filtered cigar and traditional cigar use. Because traditional cigars are often used rarely (once per week or once per month), the understanding of “fairly regular” may be different from “fairly regular” use of filtered cigars or cigarillos.

Minor correction:

Page 6, line 111 – Correct to keep commas consistent.

Reviewer #2: Understanding patterns of cigar use initiation by age, race, gender, and product type is critical to develop appropriate prevention efforts and tobacco control policies. While this is an important study that contributes valuable knowledge to the field, there are several areas that can be improved. Namely, the importance of differentiating cigar product type can be better described in the Introduction and Discussion – the results indicated that different epidemiological patterns emerged when making this distinction, so this argument should be highlighted more prominently. Additionally, some methodological approaches can be more clearly described. Below are specific recommendations to improve the manuscript.

INTRODUCTION

- The authors acknowledged an important limitation of prior studies on cigar use, which often collapse different types of cigar products into one broad category. However, the importance of studying cigar use by type was not well-established in the Introduction. Why is it so critical to investigate these differences (e.g., are there differences in health risks, addiction potential, appeal, price, etc.)? This would make a more compelling case for the study.

- Similarly, it would be helpful to briefly describe how these 3 products differ in addition to size. For example, large cigars are generally used less frequently and are not commonly inhaled, some cigarillos are primarily used as blunts to smoke marijuana, filtered cigars are often used as cheaper cigarette substitutes. This is mentioned in the Discussion, but this context seems important to describe upfront.

METHODS

- The authors created a combined category for cigarillo and filtered cigar use, noting that this is “consistent with other published studies.” The collapsing of these categories is at odds with the authors’ criticism of prior studies detailed in the introduction. Moreover, the ways in which these products are regulated and used differ: little filtered cigars, which are sometimes indistinguishable from cigarettes, are usually sold in packs of 20 and are taxed at the same rate as cigarettes at the federal level. Cigarillos, however, vary more widely in their flavor offerings, pack sizes, cost, and consumer profiles. If the data are available, it seems like a missed opportunity to not distinguish between these products.

- The last sentence in the “Ever use” section repeats what was said in the second sentence and can be removed. This is true for the “Past 30 day use” and “Fairly regular use” sections, as well.

- The description about how age of initiation was determined (i.e., adding the number of weeks between waves to the participant’s age) was unclear. Without knowing their birthdate, how is this used to determine age? For example, two participants who are 16-years old at wave 1 could be almost a year apart in age. A combination of birthdate plus dates of survey waves would be more precise. Is this information available? More clarification is needed in this section.

- Another limitation is that it is not possible to determine the age when participants actually initiated each category of use (at least in my understanding of how these measures were described). For example, if in the second wave a participant states that they are an ever user, past 30 day user, or used fairly regularly, they may have “initiated” those behaviors immediately after wave 1, at the mid-point between waves, or just before wave 2. Since they are only asked about these behaviors during their interviews – but are not asked when these behaviors actually first occurred - how can the authors determine the age when these behaviors first took place?

RESULTS

- Because of the sample size issues for the “fairly regular” use variable (i.e., unstable estimates for traditional cigar use across all ages and across most ages for the other cigar variables), it might be worth considering dropping this variable from the analyses. This might also help pare down the paper, which has a fairly wide scope.

TABLES

- The title for Table 1 seems disjointed and is a bit confusing to interpret.

- In Table 1, it might be more interpretable to present the percentage of participants who use another tobacco product, instead of the average number of products used (which are all 1).

- In Table 5, some of the numbers for “fairly regular use” seem too high. For example, the cumulative incidence by 18 years old is over 50% for most groups, which is substantially higher than ever or past 30 day use.

- Moreover, in Table 5, why are only certain categories presented? (e.g., susceptibility only for traditional cigars, fairly regular use only for filtered cigars/cigarillos)

DISCUSSION

- There is a typo on page 25, line 444. It should read “2013-2014.”

- The authors should clarify their recommendation to differentiate cigar interventions by behavior (e.g., ever use, past 30 day, “fairly regular”). Traditionally, interventions target primary prevention or cessation. What is meant by this?

- The authors focus mainly on the filtered cigar/cigarillo findings, and briefly describe findings for traditional cigars, but the observed differences between these two products – especially by race – seem noteworthy and should be discussed in greater detail. White youth are more likely to initiate traditional cigars, whereas Black youth have considerably higher rates of initiation (and at earlier ages) for filtered cigars/cigarillos. The typical use behaviors (e.g., frequency, purpose) and product characteristics (e.g., price, flavorings, packaging size, accessibility, etc.) between these two categories differ, and may influence racial/ethnic disparities in youth tobacco use. Discussing this issue might be one way to justify the importance of analyzing cigar use epidemiology by product type, which the authors set out to do in the Introduction.

6. PLOS authors have the option to publish the peer review history of their article (what does this mean?). If published, this will include your full peer review and any attached files.

Reviewer #1: No

Reviewer #2: No

---

## [Author Response · Author response to Decision Letter 0]

20 Oct 2020

See file attached with answers to the reviewers. Thank you.

---

## [Decision Letter · Decision Letter 1]

10 Nov 2020

PONE-D-20-26733R1

AGE OF INITIATION OF CIGARILLOS, FILTERED CIGARS AND/OR TRADITIONAL CIGARS AMONG YOUTH: FINDINGS FROM THE POPULATION ASSESSMENT OF TOBACCO AND HEALTH STUDY, 2013-2017

PLOS ONE

Dear Dr. Perez,

Thank you for submitting your revised manuscript to PLOS ONE. After careful consideration, we feel that it has merit, however reviewer #2  had a few additional, relatively minor issues for you to address before we can accept this for publication.  Therefore, we invite you to submit a revised version of the manuscript that addresses the points raised during the review process.

We look forward to receiving your revised manuscript.

Kind regards,

Michael Cummings, PhD

Academic Editor

PLOS ONE

Reviewers' comments:

Reviewer's Responses to Questions

**Comments to the Author**

1. If the authors have adequately addressed your comments raised in a previous round of review and you feel that this manuscript is now acceptable for publication, you may indicate that here to bypass the “Comments to the Author” section, enter your conflict of interest statement in the “Confidential to Editor” section, and submit your "Accept" recommendation.

Reviewer #1: All comments have been addressed

Reviewer #2: (No Response)

2. Is the manuscript technically sound, and do the data support the conclusions?

Reviewer #1: Yes

Reviewer #2: Yes

3. Has the statistical analysis been performed appropriately and rigorously? 

Reviewer #1: Yes

Reviewer #2: Yes

4. Have the authors made all data underlying the findings in their manuscript fully available?

Reviewer #1: Yes

Reviewer #2: No

5. Is the manuscript presented in an intelligible fashion and written in standard English?

Reviewer #1: Yes

Reviewer #2: Yes

6. Review Comments to the Author

Reviewer #1: I have no further comments to make to the authors. You have doe a great job in addressing my previous comments.

Reviewer #2: The authors did a commendable job revising this manuscript, which is considerably clearer and stronger. While the Introduction and Discussion sections are substantially improved, I still think a sentence or two could be added about why it is so important to differentiate cigars by subtype from a health perspective. The authors did a great job describing how the products look different and are used differently, but why is this important for individual/population health? One thought is that smaller cigars are generally used more frequently and are often inhaled, while traditional cigars are usually not inhaled and are used more infrequently. This could potentially influence their addictive potential or even their health risks. Two minor comments are listed below:

Pg. 4, lines 67-71: The second part of this sentence (i.e., “socializing when smoking traditional cigars was reported in 2013-2014 with the highest proportion for traditional cigar use exhibited among adults”) was unclear. The sentence would benefit from restructuring.

Pg. 7, lines 140-141: The sentence “PATH excluded ‘blunt-only’ users” makes it sound like these individuals were excluded from the PATH study. Please clarify that PATH created derived variables for cigar product use, and that blunt-only users were excluded from these variables.

7. PLOS authors have the option to publish the peer review history of their article (what does this mean?). If published, this will include your full peer review and any attached files.

Reviewer #1: No

Reviewer #2: No

---

## [Author Response · Author response to Decision Letter 1]

17 Nov 2020

AGE OF INITIATION OF CIGARILLOS, FILTERED CIGARS AND/OR TRADITIONAL CIGARS AMONG YOUTH: FINDINGS FROM THE POPULATION ASSESSMENT OF TOBACCO AND HEALTH STUDY, 2013-2017

We want to thank the reviewers for the excellent comments which help us to make this paper stronger. Our answers are below after each comment. 

Have the authors made all data underlying the findings in their manuscript fully available?

Reviewer #2: No

Response: We clarified that the PATH data we used for all analyses are restricted data from a third party. Here we include the data availability statement we made for the paper. Interested researchers can apply for the restricted data from the third party directly.

Data Availability Statement: All the data from waves 1-4 are available from the Population Assessment of Tobacco and Health (PATH) Study [United States] Restricted-Use Files. Inter-university Consortium for Political and Social Research [distributor], 2020-06-24. https://doi.org/10.3886/ICPSR36231.v25.

Data are available from https://www.icpsr.umich.edu/web/NAHDAP/studies/36231

This is in reference #18 in the current version of the manuscript. The website is shown as a footnote in all tables to highly the location of the data. We believe with this we comply with the journal policy.

Reviewer #2: 

The authors did a commendable job revising this manuscript, which is considerably clearer and stronger. While the Introduction and Discussion sections are substantially improved, I still think a sentence or two could be added about why it is so important to differentiate cigars by subtype from a health perspective. The authors did a great job describing how the products look different and are used differently, but why is this important for individual/population health? One thought is that smaller cigars are generally used more frequently and are often inhaled, while traditional cigars are usually not inhaled and are used more infrequently. This could potentially influence their addictive potential or even their health risks. Two minor comments are listed below:

Response: We thank the reviewer for the excellent comments. Based on your comments, we added the following sentence in lines 79-82 for further clarification:

“From the health perspective, cigarillos or filtered cigars are generally used more frequently in youth and are often inhaled, which may increase young smokers’ risk for addiction to nicotine and/or poor health outcomes[10, 11]. However, the traditional cigars are usually not inhaled and are used less frequently in youth.”

Pg. 4, lines 67-71: The second part of this sentence (i.e., “socializing when smoking traditional cigars was reported in 2013-2014 with the highest proportion for traditional cigar use exhibited among adults”) was unclear. The sentence would benefit from restructuring.

Response: We have restructured this sentence to make it clear.

Pg. 7, lines 140-141: The sentence “PATH excluded ‘blunt-only’ users” makes it sound like these individuals were excluded from the PATH study. Please clarify that PATH created derived variables for cigar product use, and that blunt-only users were excluded from these variables.

Response: Yes, we agree with the reviewer and we clarified this sentence based on this suggestion. 

Thank you

---

## [Editor Report · Decision Letter 2]

20 Nov 2020

AGE OF INITIATION OF CIGARILLOS, FILTERED CIGARS AND/OR TRADITIONAL CIGARS AMONG YOUTH: FINDINGS FROM THE POPULATION ASSESSMENT OF TOBACCO AND HEALTH STUDY, 2013-2017

PONE-D-20-26733R2

Dear Dr. Perez,

We’re pleased to inform you that your manuscript has been judged scientifically suitable for publication and will be formally accepted for publication once it meets all outstanding technical requirements.

Kind regards,

Michael Cummings, PhD

Academic Editor

PLOS ONE
---

## [Editor Report · Acceptance letter]

24 Nov 2020

PONE-D-20-26733R2 

Age of initiation of cigarillos, filtered cigars and/or traditional cigars among youth: findings from the Population Assessment of Tobacco and Health (PATH) study, 2013-2017. 

Dear Dr. Pérez:

I'm pleased to inform you that your manuscript has been deemed suitable for publication in PLOS ONE. Congratulations! Your manuscript is now with our production department. 

Kind regards, 

on behalf of

Dr. Michael Cummings 

Academic Editor

PLOS ONE